# Formation of Kinetics Coherent Structures in Weakly Collisional Media

**Alexander Karimov [1,2,3,*,†] and Vladislav Bogdanov [1,†]**

[1] Department of Electrophysical Facilities, National Research Nuclear University MEPhI, Kashirskoye hwy, 31, Moscow 115409, Russia; bvk001@campus.mephi.ru

[2] Institute for High Temperatures, Russian Academy of Sciences, Izhorskaya 13/19, Moscow 127412, Russia

[3] Research Institute of Quality, Safety and Technologies of Specialized Food Products of the PRUE G.V. Plekhanov, Stremyanny Lane, 36, Building 6, Moscow 117997, Russia

* Correspondence: arkarimov@mephi.ru; Tel.: +7-903-564-0123

† These authors contributed equally to this work.

**Abstract:** The formation of nonlinear, nonstationary structures in weakly collisional media with collective interactions are investigated analytically within the framework of the kinetic description. This issue is considered in one-dimensional geometry using collision integral in the Bhatnagar-Gross-Krook form and some model forms of the interparticle interaction potentials that ensure the finiteness of the energy and momentum of the systems under consideration. As such potentials, we select the Yukawa potential, the $\delta$-potential, which describes coherent structures in a plasma. For such potentials we obtained a dispersion relation which makes it possible to estimate the size and type of the forming structures.

**Keywords:** plasma; physical kinetics; Bhatnagar-Gross-Krook collision integral; Vlasov equations; coherent structures

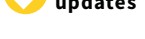



## 1. Introduction

There are a lot of natural systems which are capable of exchanging energy and negentropy with the immediate environment [1,2]. In this case, the steady time-dependent states of the systems appear to be possible as a result of two effects: the formation of coherent states due to the self-consistent potential field and the formation of dissipative equilibria due to collisions. A well-known example of dissipative processes is the Beloysov-Ghabotinskyl reaction [2,3], whereas the formation of nonlinear wave-like vortex-like structures in the collisionless Coulomb plasmas [4,5] is the most vivid and vital sample of the system with the purely potential interaction between particles, which have been described by soliton-like distributions. In this case, one should keep in mind that such wave-like solutions possess some properties of particles [6,7]. For example, the polar molecules clouds can combine into hyperparticle structures which one may consider to be macroscopic atoms.

From this perspective, we could suggest that the internal structure as well as the physical properties of such hyperparticle samples is similar to real atoms. In particular, we may expect that they are capable of accumulating some power inside themselves and to produce (gate out) it during the fusion. It is clear that such macroscopic entities can be formed in various systems with collective interacting particles. (We shall call such media as plasma-like media.)

Therefore, it would be interesting to study the possibility of such macroscopic, time-dependent entities formation in weakly dissipative media with different collective interaction between particles. Therefore, the special question within this study is dependence of coherent states on the type of potential – Lennard-Jones, Yukawa, Coulomb potential, etc. To understand this, we are going to consider the influence of the potential type on the formation of coherent states in ideal, weakly collision plasma-like media.

This article discusses an approach to describing the behavior of particles in the approximation of physical kinetics. We examined the formation of coherent structures in the environment, consisting of a Maxwellian ensemble of particles. At the same time, various potentials of interaction between particles are considered.

## 2. Basic Equations

The medium is assumed to be infinite, i.e., the pure initial value problem is to be studied. Within the framework of a kinetic approach, the governed equations for the one-particle distribution function $f = f(t, \mathbf{r}, \mathbf{v})$ describing polar molecules dynamics are

$$\frac{\partial f}{\partial t} + \mathbf{v} \cdot \nabla f - \frac{\mathbf{F}}{m} \cdot \frac{\partial f}{\partial \mathbf{v}} = I_c, \tag{1}$$

where $\mathbf{F}$ is the force influencing the medium particles of mass $m$ and $I_c$ is the collision integral.

To show the basic features of formation of coherent structures in collisionless and weakly collision media we start from the simplest possible form of collision integral

$$I_c = -\nu(f - f_e) \tag{2}$$

where $\nu$ is the collision frequency and $f_e$ is some equilibrium distribution to be determined.

Here we suppose that the interaction between particles depends only on distance between particles $|\mathbf{r} - \mathbf{r}'|$ but not their velocities $\mathbf{v}$ and $\mathbf{v}'$. So one can represent self-consistent force $\mathbf{F}$ in terms of the scalar potential $\Phi = \Phi(t, \mathbf{r})$,

$$\mathbf{F} = -\nabla \Phi,$$

where potential is defined by the relation [4]:

$$\Phi(t, \mathbf{r}) = \int K_{1,2}(|\mathbf{r} - \mathbf{r}'|) f(t, \mathbf{r}', \mathbf{v}') d\mathbf{r}' dv' + \tag{3}$$

$$+ \int K_{1,2,3}(|\mathbf{r} - \mathbf{r}'|, |\mathbf{r} - \mathbf{r}''|, |\mathbf{r}' - \mathbf{r}''|) f(t, \mathbf{r}', \mathbf{v}') f(t, \mathbf{r}'', \mathbf{v}'') d\mathbf{r}' dv' d\mathbf{r}'' dv'' + \cdots + \Pi(t, \mathbf{r}),$$

where $\Pi(t, \mathbf{r})$ is some known function. In the general case, this function reflects the influence of external superposed factors.

Then, let us discuss the influence of separate terms on the nonlinear properties of the system; however, now for simplicity we restrict our study to the case when

$$\Phi(t, \mathbf{r}) = \int K(|\mathbf{r} - \mathbf{r}'|) f(t, \mathbf{r}', \mathbf{v}') d\mathbf{r}' dv' + \Pi(t, \mathbf{r}). \tag{4}$$

The kernel of integral relation is determined by the nature of interaction between particles of the system, for example, that may be dipolar interaction. However, the present approach is valid only when the kernel $K(\mathbf{r}, \mathbf{r}')$ satisfies the relation

$$\gamma = \int_{-\infty}^{+\infty} K(|\mathbf{r} - \mathbf{r}'|) d\mathbf{r}' < \infty, \tag{5}$$

whereas Coulomb-type potentials cannot be defined by such way since for these fields the relation (5) is violated. In this case we have to use Poison equation.

We are going to construct some partial solution of the issue (1), (4) using [8]:

$$f(t, \mathbf{r}, \mathbf{v}) = \sum_{k=0}^{L} F_k(t, \mathbf{v}) \Phi^k(t, \mathbf{r}), \tag{6}$$

where $F_k(t, \mathbf{v})$ are unknown functions to be determined, and $\Phi(t, \mathbf{r})$ is determined by relation (4). We assume that $L$ is a finite integer or infinite value. We are going to seek the local equilibrium distribution function in similar form

$$f_e(t, \mathbf{r}, \mathbf{v}) = \sum_{k=0}^{L} C_k F_0(t, \mathbf{v}) \Phi^k, \tag{7}$$

where $C_k$ are some constants.

Additionally, we suppose that $\Phi(t, \mathbf{r})$ satisfies

$$\frac{\partial \Phi}{\partial t} + \mathbf{V} \cdot \nabla \Phi = 0, \tag{8}$$

where drift velocity $\mathbf{V}$ is a constant.

By applying (6)–(8) to Equation (1), we find

$$\sum_{k=0}^{L} \left\{ \frac{\partial F_k}{\partial t} + \nu(F_k - C_k F_0) + (k+1)F_{k+1}(\mathbf{v} - \mathbf{V}) \cdot \nabla \Phi + \nabla_v F_k \cdot \nabla \Phi \right\} \Phi^k = 0. \tag{9}$$

If there is no connection between $\Phi(t, \mathbf{r})$ and $F_k(t, \mathbf{v})$, relation (9) has to be satisfied for any power of $\partial \Phi / \partial q_j$ and $\Phi$ (here $q_1 = x$, $q_2 = y$, $q_3 = z$). So we obtain

$$\nabla_v F_k + (k+1)F_{k+1}(\mathbf{v} - \mathbf{V}) = 0, \tag{10}$$

$$\frac{\partial F_k}{\partial t} = -\nu(F_k - C_k F_0). \tag{11}$$

The solution of (11) can be written as

$$F_k(t, \mathbf{v}) = W_k(\mathbf{v}) \exp(-\nu t) + C_k F_0(\mathbf{v}), \tag{12}$$

where $W_k(\mathbf{v})$ is some functions. Then substituting (12) into Equation (10) we have

$$(k+1)(\mathbf{v} - \mathbf{V})\left(W_{k+1}e^{-\nu t} + C_{k+1}F_0\right) + C_k \nabla_v F_0 + e^{-\nu t}\nabla_v W_k = 0 \tag{13}$$

from which we can obtain

$$(k+1)C_{k+1}F_0(\mathbf{v} - \mathbf{V}) + C_k \nabla_v F_0 = 0, \tag{14}$$

$$(k+1)W_{k+1}(\mathbf{v} - \mathbf{V}) + \nabla_v W_k = 0. \tag{15}$$

For

$$C_{k+1} = \frac{C_k}{(k+1)T},$$

where $T > 0$ is a constant which can be considered to be temperature; from Equation (14) it follows

$$F_0 = C_0 \exp\left[-\frac{(\mathbf{v} - \mathbf{V})^2}{2T}\right], \tag{16}$$

where $C_0$ is an arbitrary constant. From (15) we obtain

$$W_k = \frac{1}{k!}D^k W_0(v), \quad k = 1, 2, \ldots, L, \tag{17}$$

where

$$D = \frac{\mathbf{V} - \mathbf{v}}{(\mathbf{v} - \mathbf{V})^2}\nabla_v.$$

Finally, the substitution of (16) and (17) into (12) leads to

$$F(t, \mathbf{v}) = \frac{e^{-\nu t}}{k!} D^k W_0(\mathbf{v}) + \frac{C_0}{k!} \exp\left[-\frac{(\mathbf{v} - \mathbf{V})^2}{2T}\right] \tag{18}$$

and the distribution function can then be written as

$$f(t, \mathbf{r}, \mathbf{v}) = \sum_{k=0}^{L}\left[e^{-\nu t} D^k W_0(\mathbf{v}) + \frac{C_0}{T^k} e^{-(\mathbf{v}-\mathbf{V})^2/2T}\right]\frac{\Phi^k}{k!}. \tag{19}$$

The function $W_0(\mathbf{v})$ can be any function of velocity $\mathbf{v}$ such that all moments of the distribution function (19),

$$I_n(t, \mathbf{r}) = \int_{-\infty}^{+\infty} \mathbf{v}^n f(t, \mathbf{r}, \mathbf{v}) d\mathbf{v},$$

must be finite, i.e.,

$$|I_n(t, \mathbf{r})| < \infty, \quad n = 0, 1, 2, \ldots. \tag{20}$$

and for this kernel $K(\mathbf{r}, \mathbf{r}')$ provide the finite $\Phi(t, \mathbf{r})$, i.e.,

$$\int K(|\mathbf{r} - \mathbf{r}'|) f(t, \mathbf{r}', \mathbf{v}') d\mathbf{r}' d\mathbf{v}' < \infty. \tag{21}$$

Moreover, we have to require

$$D_k = \int_{-\infty}^{+\infty} D^k W_0(\mathbf{v}) d\mathbf{v} < \infty. \tag{22}$$

It is essential to stress conditions (20)–(22) determine the allowable choices of $W_0(\mathbf{v})$. To define the admissible form of $\Phi(t, \mathbf{r})$ we put (19) into (6). As a result we obtain

$$\Phi(t, \mathbf{r}) = \sum_{k=0}^{L} \int_{-\infty}^{+\infty} \frac{K(|\mathbf{r} - \mathbf{r}'|)}{k!}\left[C + D_k e^{-\nu t}\right]\Phi^k(t, \mathbf{r}') d\mathbf{r}'. \tag{23}$$

Here we used $C_0 = C/(2\pi T)^{3/2}$.

Relations (23) and (19) with conditions (20)–(22) form the basis for most of the following analysis.

## 3. Maxwellian Type Distributions

For simplicity, we shall analyze Equation (23) in the case when $L \to \infty$ and when $F_0$ belongs the class of Maxwellian functions,

$$F_0(\mathbf{v}) = \frac{1}{(2\pi\theta)^{3/2}} \exp\left[-\frac{(\mathbf{v} - \mathbf{V})^2}{2\theta}\right], \tag{24}$$

where $\theta > 0$ is a constant which plays role of second temperature. It should be noted that in general case we can suppose $T \neq \theta$. Then from (19) and (24) it follows

$$f(t, \mathbf{r}, \mathbf{v}) = \frac{1}{(2\pi\theta)^{3/2}} e^{-\nu t} \exp\left[-\frac{(\mathbf{v} - \mathbf{V})^2 +}{2\theta} + \frac{\Phi}{\theta}\right] + \frac{1}{(2\pi T)^{3/2}} \exp\left[-\frac{(\mathbf{v} - \mathbf{V})^2}{2T} + \frac{\Phi}{\theta}\right] \tag{25}$$

In this case, instead of (23) we obtain

$$\Phi(t, \mathbf{r}) = e^{-\nu t} \int_{-\infty}^{+\infty} K(|\mathbf{r} - \mathbf{r}'|) \exp(\Phi/\theta) d\mathbf{r}' + C \int_{-\infty}^{+\infty} K(|\mathbf{r} - \mathbf{r}'|) \exp(\Phi/T) d\mathbf{r}' \tag{26}$$

The nonlinear equation (26) admits spatially uniform solution $\Phi_0 = \Phi_0(t)$ which is determined from

$$\Phi_0 = \gamma\left(e^{-\nu t + \Phi_0/\theta} + Ce^{\Phi_0/T}\right), \tag{27}$$

where $\gamma$ is determined by (5). There is its nontrivial solution. In particular, if we set $\theta = T$ and $\Phi_0/\theta \ll 1$, one can obtain

$$\Phi_0 = \frac{\gamma\left(e^{-\nu t} + C\right)}{1 - C(e^{-\nu t} + C)/\theta}. \tag{28}$$

Let us consider (28) as a state of local equilibrium, near which the basic equation is linearized, for this we substitute

$$\Phi = \Phi_0(t) + \delta\Phi_0(t, \mathbf{r})$$

into (26). As result we have

$$\delta\Phi - \frac{e^{\Phi_0/\theta}}{\theta}\left(e^{-\nu t} + C\right)\int_{-\infty}^{+\infty} K(|\,\mathbf{r} - \mathbf{r}'\,|)\delta\Phi(\mathbf{r}')d\mathbf{r}' = 0. \tag{29}$$

Let us analyze the behavior of Equation (28) with respect to different values of handling parameters $\nu$, $\theta$. In particular, in the limit $\nu t \to \infty$, Equation (29) admits stationary solution of the form

$$\delta\Phi = Ae^{i\mathbf{k}\mathbf{r}}. \tag{30}$$

Substituting (30) into the (29), we obtain

$$1 - \lambda \int_{-\infty}^{+\infty} K(|\,\mathbf{r} - \mathbf{r}'\,|)\exp(-i\mathbf{k}(\mathbf{r} - \mathbf{r}')d\mathbf{r}' = 0. \tag{31}$$

where

$$\lambda(\theta, C) = \frac{C\exp(\Phi_0/\theta)}{\theta}. \tag{32}$$

One can estimate the integral

$$\int_{-\infty}^{+\infty} K(|\,\mathbf{r} - \mathbf{r}'\,|)\exp(-i\mathbf{k}(\mathbf{r} - \mathbf{r}')d\mathbf{r}' = \int_0^\infty \int_0^\pi \int_0^{2\pi} K(\rho)\exp[-ik\rho\cos(\vartheta)]\rho^2\sin(\vartheta)d\varphi d\vartheta d\rho$$

$$= \frac{4\pi}{k}\int_0^{+\infty} K(\rho)\rho\sin(k\rho)d\rho.$$

Thus, we obtain the following dispersion relation

$$1 - \frac{4\pi\lambda}{k}\int_0^{+\infty} K(\rho)\rho\sin(k\rho)d\rho = 0. \tag{33}$$

which defines the existence of spatial wavelet-structures for the real values $k$. Equation (33) has nontrivial, real root $k$ if and only if the integral in (33) is negative. Such behavior is possible owing to the initial conditions (see Equation (32)) or the kernel of interaction $K(\rho)$.

## 4. Different Types of Kernels

As an illustration, we now calculate the roots of Equation (33) for the kernel of type

$$K(\mathbf{r}, \mathbf{r}') = \frac{\mu}{|\,\mathbf{r} - \mathbf{r}'\,|}\exp\left(-q\,|\,\mathbf{r} - \mathbf{r}'\,|\right), \tag{34}$$

where $\mu$ and $q$ are some constants depending on the kind of the interactions which admits elementary analytical consideration and has some physical meaning. Inserting (34) into (33) we obtain

$$1 - \frac{4\pi\lambda\mu}{k^2 + q^2} = 0. \tag{35}$$

It is evident that this equation has real roots in the case

$$4\pi\lambda\mu \geq q^2 = 0. \tag{36}$$

The spatial length of periodic structures

$$k^{-1} = (4\pi\lambda\mu - q^2)^{-1/2} \tag{37}$$

is defined by the initial conditions over constant $\lambda$ and the parameters of interaction $q$ and $\mu$. This example shows that for some initial conditions and kernels one can expect the formation of spatially nonuniform structures from initial uniform states [9]. The relation (33) or (35) determines only value of **k** but its direction is unknown. Using (37) we can determine the size of the vortex depending on the initial parameters of the system.

As another result of the application of this method, we give an example of the formation of coherent structures in the "billiard balls" model. In that case, the kernel of interaction is shown by

$$K(\mathbf{r}, \mathbf{r}') = \mu\delta(\mathbf{r} - \mathbf{r}' - \mathbf{a}) \tag{38}$$

where $\mu$ is some constant depending of the kind of interactions and $|\mathbf{a}|$ is the radius of every particle of the media. Inserting (38) into dispersion relation (33) we obtain

$$1 - \frac{4\pi\lambda\mu a \sin(ak)}{k} = 0. \tag{39}$$

We can transform Equation (39) into a more convenient form

$$k = 4\pi\lambda\mu a \sin(ak). \tag{40}$$

Equation (40) has some sets of solutions. This shows us that if we imagine that particles of a medium interact only through collisions, coherent vortexes still appear. They can have different sizes that correspond to different positive roots of the Equation (40). Additionally, we add that if

$$q >> 1, \tag{41}$$

where $q$ is the constant included in (34), the kernel (34) is greatly approximated by a delta-function such as (38). That is, the Yukawa kernel model under the condition of a low intensity of particle interaction (41) could be approximated by the model of billiard balls.

## 5. Discussion and Conclusions

To describe the formation of kinetic coherent structures in weakly collisional medium with collective interactions we used one-dimensional, time-dependent nonlinear Vlasov equations with simplest collision integral in the Bhatnagar-Gross-Krook form. Additionally, in our model of collisional medium we have used some physically admissible forms of the interparticle interaction potentials providing the finiteness of the energy and momentum of the systems under consideration. As such potentials, we select the Yukawa potential, the $\delta$-potential, and the soliton potential, which may describe electron holes in a plasma [10].

We applied the method of constructing time-dependent solutions of the kinetics equations based on the expansion of the distribution function as a series in positive powers of the interparticle interaction potentials [8]. The key point of this method consists in the representation of nonstationary solution and local equilibrium distribution function in similar form (see Equations (6) and (7)). As a result, we managed to show that a Maxwellian time-dependent distribution function (19) appears as a natural solution of the considered

initial-value problem (see Equations (10) and (11)) when such Maxwellian time-dependent distribution can be formed for any interparticle interaction potential.

In this class of Maxwellian distributions for the considered potentials, we obtained a dispersion relation Equation (33) which makes it possible to estimate the size and type of the forming structures in locally equilibrium media. Analyzing dispersion relation (33), we presented coherent structures as wave solutions of the equation, which describes variations with respect to the equilibrium states of system. Such an approximation is valid if the spatial scales of the resulting structures are small with respect to the characteristic scale of the system [11].

Analyzing relation (33), we presented coherent structures as wave solutions of equation, which describes variations with respect to the equilibrium states of system [12]. Taking into account such high sensitivity of our pattern to perturbations in the initial conditions, one can come to the conclusion that there exists a small interval in the system parameters in which it is possible to see the effect described in reality. Our consideration was limited to the case of Maxwellian distributions. In this regard it would be interesting to study the Lorenz-Fermi distributions which may be more relevant for some natural cases.

**Author Contributions:** Conceptualization, A.K., V.B.; methodology, A.K.; validation, A.K., V.B.; investigation, V.B., writing review and editing, V.B.; supervision, A.K.; funding acquisition, A.K. All authors have read and agreed to the published version of the manuscript

**Funding:** The research has been supported by the grant "Governing for the processes of structure formation in high-molecular compounds under non-equilibrium conditions" of Plekhanov Russian University of Economics.

**Institutional Review Board Statement:** Not applicable.

**Informed Consent Statement:** Not applicable.

**Data Availability Statement:** Not applicable.

**Acknowledgments:** The authors would like to thank Hans Schamel for his support in developing the theory. Specifically, for providing information and access to existing works on the topic, as well as for verifying the truth of the methods presented in the article.

**Conflicts of Interest:** The authors declare no conflict of interest.

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
