# Peer review of "Formation of Kinetics Coherent Structures in Weakly Collisional Media"

_plasma, doi:10.3390/plasma4020024_

Round 1
Reviewer 1 Report
Referee report
Formation of Kinetics Coherent Structures in Weak Collision Mediaby A. Karimov and V. Bogdanov.
This manuscript constructs a nonlinear solution to the kinetic equation which describes coherent structures in a plasma. The novel point is to account for weak, but qualitatively important collisions.
The analysis is sound and correct, and the assumptions are clear and reasonable. I will recommend it for publication in Plasmas, provided the following points are adequately adressed.
1.The introduction is very general, which is good for a first part, but it misses a second part to give the more specific context of the problem at hand, which has been studied analytically and numerically by various authors in the literature. See papers by Dudkovskaia, others by Lesur, and others by Lilley for the 1D problem, including phase-space holes (Schamel is already cited) and BGK-like phase-space vortices ; and papers by Kosuga for a quasi-2D context. Then the results of the present manuscript may also be linked with the latter results in the literature.
2. Page 2, typo: ”i,e. i.e.”
3. The English formulation is quite strange or grammatically incorrect at several places in the manuscript, and in particular the sentence on top of page 5.
4. The following is only an advice, which the authors may choose to ignore. The paper could improve its impact by providing simple numerical examples, and illustrating figures.
Author Response
We have read your comments and made corrections to the introduction text, as well as corrected errors throughout the article. Thank you very much for your support in its implementation.

Reviewer 2 Report
An increased understanding of the formation of coherent structures in plasmas is most important, and deserves additional attention. The present paper adds a new important investigation in this field. I have checked the authors deivations, and concluded that they are correct.
Before publication, however, I suggest the following minor corrections:
- In the title It seems to me that the words "Weak Collision" may be replaced by "Weakly Collisional".
- In the Introduction I suggest that "Lennart-Jons" is replaced by "Lennard-Jones".
- In Ref. 3, "Oscilations" should be replaced by "Oscillations".
- Maybe the authors could in their Section 5 mention that it may be possible in forthcoming investigations to extend their analysis also to rotating plasmas (see e.g. A.R. Karimov et al., Journal of Physics: Conference Series 747, 012077 (2016) , as well as related papers)
Author Response

(The authors gave the same response as above.)
